# Moral Sensitivity and Emotional Intelligence in Intensive Care Unit Nurses

**DOI:** 10.3390/ijerph19095132

**Published:** 2022-04-23

**Authors:** Biyun Ye, Esther Luo, Jie Zhang, Xuelei Chen, Jingping Zhang

**Affiliations:** 1Xiangya School of Nursing, Central South University, Changsha 410013, China; ye_by2022@csu.edu.cn (B.Y.); zzuzhangjie@163.com (J.Z.); chenxl2020@csu.edu.cn (X.C.); 2School of Public Health, Yale University, New Haven, CT 06510, USA; esther.luo@yale.edu

**Keywords:** intensive care unit, nurses, emotional intelligence, moral sensitivity

## Abstract

Moral sensitivity helps individuals resolve moral dilemmas as a precursor to moral decision-making. Intensive care unit (ICU) nurses are at high risk for encountering moral dilemmas and should have the moral sensitivity to recognize moral issues. The activities of ICU nurses in moral decision-making are guided by moral sensitivity but are also based on emotional intelligence (EI). EI, be recognized as an integral part of moral sensitivity with long-standing theoretical foundations. It is necessary to explicate the true role of EI in moral sensitivity through empirical research. To measure the level of moral sensitivity of ICU nurses and determine the relationship between moral sensitivity and EI. We recruited 467 ICU nurses of ten hospitals from March to June 2021 in Hunan Province, China for a cross-sectional questionnaire survey. The moral sensitivity and EI were measured using the Moral Sensitivity Questionnaire-Revised Version into Chinese (MSQ-R-CV) and the Wong and Law Emotional Intelligence Scale-Version into Chinese (WLEIS-C). A self-report questionnaire covered sociodemographic characteristics. The average moral sensitivity score of ICU nurses was 39.41 ± 7.21. The average EI score was significantly positively correlated with the moral sensitivity score (*p* < 0.001). This study demonstrated that the moral sensitivities of ICU nurses were at medium levels. EI of ICU nurses can indeed affect their moral sensitivity, and the impact of each element of EI should be clarified for practical application.

## 1. Introduction

Ethical and moral concerns are embedded in modern nursing practices [1]. Several documents show that nursing staff often experience moral conflicts, which are antecedents to moral distress [2,3,4,5]. A moral dilemma refers to the psychological imbalance and negative feelings experienced when a nurse cannot follow through with what he or she believes to be moral actions, thereby compromising his or her professional integrity [6]. Such dilemmas are root causes of psychological distress and burnout and may have serious consequences: increased moral pressure, decreased job satisfaction and quality of care, and, in some cases, a nurse’s resignation [7,8,9,10,11]. Moral dilemmas are particularly salient in intensive care units (ICUs). As one of the most critical and tense departments in a hospital, the ICU is staffed by specialists who care for patients with critical, life-threatening conditions. Janvier et al. [12] report that the level of moral dilemmas is higher in ICUs than in other departments. As a result, ICU nurses are at a higher risk for facing moral dilemmas due to the moral conflicts created by pressures in these settings. These pressures stem from technological advances, high-intensity work environments, patients’ and their families’ high expectations, futility of further treatment, ineffective pain management, and frequent exposure to death [1,13,14]. Efficient moral decision-making can help nurses cope with moral conflicts and effectively resolve moral dilemmas [15,16].

As the first component of Rest’s Four-Components Model (FCM) [17,18], moral sensitivity is a necessary precursor for moral decision-making and behavior. Although the literature at times uses the terms ethical sensitivity and moral sensitivity interchangeably, there is a clear distinction: Ethical sensitivity refers to the knowledge of ethics theory and principles (knowing “what” to do), while moral sensitivity relates to personal agency within interpersonal relationships (knowing “how” to do something) [19]. Moral sensitivity implies that personal qualities are important in the decision-making process. Moral sensitivity is the capacity with which nurses identify moral issues and also provide insight into their own responsibilities and the consequences of making decisions on behalf of patients [20]. Moral sensitivity enables nurses to identify patients’ needs and provide care based on the moral values nurses possess. Moreover, moral sensitivity is necessary for recognizing moral dilemmas and evaluating the advantages and disadvantages of different decisions [21,22].

When people experience moral situations in reality, they are usually triggered first by strong emotions before complex cognitive coding. Emotions can occur instantaneously without a careful understanding of the truth. Emotions can stimulate and guide moral cognition and affect people’s moral sensitivity together [17]. In the absence of emotion, the moral principles and norms are just rigid and virtual moral symbols, which cannot penetrate deep into people’s hearts and internalize and reconstruct the existence of the subject itself.

Emotions have solid theoretical foundations as an integral part of moral sensitivity: the interpretation of each component in Rest’s [17] four-component model of moral psychology involves complex interactions between emotion and cognition; Hoffman’s [23] study of empathy and moral development; British moral philosophers in the 17th century put forward the view that “moral differences stem from emotion rather than reason”, and they are called “emotionalists” [24]; the establishment of the emotionalist moralism of the Scottish philosopher Hume [25] in the 18th century; and the Swedish ethicist Kim Lützén [20] affirms that “moral sensitivity is not just an “emotion” (emotions can be used to identify moral conflicts). Some studies suggest that emotion is essential for moral behavior from the perspective of neurophysiological mechanisms. Damasio et al. [26] examined patients with damage to the ventromedial prefrontal cortex (the part of the brain that controls emotion). Their reasoning ability to solve moral dilemmas is no different from that of ordinary people, but they perform poorly in real situations, such as making wrong decisions and being indecisive, and even behaving irrationally. Greene et al. [27] used functional magnetic resonance imaging technology to find that in different moral dilemmas, subjects showed significant fluctuations in the activity of the anterior cingulate cortex, a brain region related to rapid emotional response. It shows that the change of moral situation will affect individual’s moral cognitive activities by triggering different degrees of emotional reaction.

However, the main moral decision models in the literature generally recommend that clinicians strip emotion from the reasoning process [28] in order to reduce bias and framing effects [29] and over-reliance on intuitive judgment [30]. Contrary to the assumption that moral decision-making should be emotionless, clinical reasoning and decisions are often made in contexts that are emotionally challenging and require nurses to actively manage their own and others’ emotions [31,32]. In seeking to understand emotional engagement in nursing practice, scholars note that the value of emotions can be defined by using the EI framework [33]. EI is defined as the ability of individuals to monitor their own and others’ emotions, distinguish between positive and negative effects of emotions, and use emotion-related information to guide their cognitive reasoning and take actions [34]. Through the review and synthesis of empirical evidence, it was found that there is enough evidence to conclude that EI of medical personnel can and does affect their reasoning and decision-making [31]. The nursing literature emphasizes the emotional aspects of clinical decision-making (CDM), and literature about end of life and elder care has especially focused on the application of emotion-influenced CDM [15]. A literature review [35] shows that many authors associate EI with CDM, indicating that emotion may be an integral part of CDM. In the literature, a link has been established between EI and clinical reasoning. The lack of quantitative research on the relationship between moral sensitivity as an ethical aspect of clinical reasoning and EI is of particular concern, especially in the professional context of ICU nurses. Studies (e.g., [36]) have demonstrated that an ICU nurse can act as a catalyst for family inclusion in moral decision-making and arbitrate between family and medical officers within the moral decision-making process. EI skills are the basis for these activities, while moral sensitivity guides them. It is necessary to explicate the role of EI in moral sensitivity in more depth in order to integrate the recognition of the true roles of EI skills in moral sensitivity. As EI contributes to moral decision-making in nursing, fully incorporating EI can improve patient safety and quality of care. Thus, we wanted to assess the levels of moral sensitivity among ICU nurses and explore any association between EI and moral sensitivity.

## 2. Methods

### 2.1. Study Design

This was a quantitative descriptive cross-sectional study.

### 2.2. Participants

From March 2021 to June 2021, we used cluster sampling to select all registered nurses in ICUs at eight tertiary hospitals and two secondary hospitals in Hunan Province, China. The inclusion criteria were: (1) registered nurses who have worked in the ICU for six months or more; (2) no history of neurological or psychological impairments; (3) have not participated in similar research studies before. Exclusion criteria included: (1) nurses who are in training, rotation, or internship; (2) nurses who did not work in the hospital during the study period for various reasons. This study required a two-sided test α = 0.05. According to the study of Huang et al. [37], the standard deviation of the total score of Chinese nurses MSQ-R-CV is σ = 7.08, and the allowable error δ = 1. The sample size N = 193 was calculated by using PASS 16.0 software. Generally speaking, the design efficiency value of cluster sampling is large, and most experts suggest 2. Therefore, the sample size required is estimated at 386. Anticipating a 10 percent failure rate (for example, some respondents may have worked in an ICU for fewer than six months), we set the sample size in this study at 425.

We contacted the director of the nursing department or the head nurse in the ICU in cooperating hospitals to obtain written informed consent, and we conducted cluster sampling of ICU nurses in the department with their assistance. Data were collected by sending online questionnaire links within WeChat or DingDing. WeChat and DingDing are free mobile office platforms used for communication and collaboration in Chinese enterprises, allowing for efficient and secure internal communication. Each has a confirmation function (i.e., a return receipt) for information delivery that can count participants’ responses and stop collecting information when all have confirmed the information delivery. The first page of the questionnaire is an informed consent page, and nurses can freely choose to participate or withdraw after confirming the information delivery.

### 2.3. Research Tools

#### 2.3.1. Self-Report Questionnaires

Questionnaires included a total of 13 variables: gender, age, religious beliefs, marital status, number of children, hospital level, working years in the ICU, average number of night shifts per month, educational level, professional category, work intensity, and whether the respondent had received ethics courses at their university, in the hospital, or in the nursing department.

#### 2.3.2. Wong and Law Emotional Intelligence Scale-Version into Chinese (WLEIS-C)

The Wong and Law [38] Emotional Intelligence Scale was developed based on Salovey and Mayer’s definition of EI and Gross’s emotion regulation theory. The scale is targeted at adult occupational groups and has a good predictive effect on workplace-related variables. The WLEIS-C is a self-administered questionnaire using a 7-point Likert scale (1 = “I totally disagree” to 7 = “I totally agree”). The scale’s score ranges from 7 to 112 points: the higher the score, the higher the EI. The scale consists of 16 items on four dimensions: self-emotional appraisal (SEA), other-emotional appraisal (OEA), regulation of emotion (ROE), and use of emotion (UOE). Wong and Law applied the scale to university students in Hong Kong and measured the correlation coefficients of the four dimensions: SEA (0.89), OEA (0.89), ROE (0.89), and UOE (0.80). The reliability and validity of the scale was good. Measured results by Wang Yefei [39] in three representative adult groups—university students, civil servants, and enterprise employees—showed that the internal consistency reliability and split half reliability of the total scale were both above 0.80. These studies show that the Chinese version of the emotional intelligence scale has good stability and reliability among Chinese adults and can measure their EI effectively. The Cronbach’s alpha coefficient of the scale obtained from the sample of this study was 0.964, and the Cronbach’s alpha coefficient of the four dimensions was SEA (0.876), OEA (0.893), ROE (0.931), and UOE (0.898).

#### 2.3.3. Moral Sensitivity Questionnaire-Revised Version into Chinese (MSQ-R-CV)

Lützén et al. [19] updated and revised the Moral Sensitivity Questionnaire (MSQ) in 2006. The Moral Sensitivity Questionnaire-Revised Version (MSQ-R) is a self-administered questionnaire designed as a 6-point Likert scale, where 1 = “I completely disagree” and 6 = “I completely agree”. The MSQ-R consists of nine items representing the three main elements of moral sensitivity: moral burden (4 items), moral strength (3 items) and moral responsibility (2 items). The composite score for the scale ranges from 9 to 54. A higher total score indicates that a nurse possesses greater moral sensitivity; specifically, a total score of <32 indicates low sensitivity, 32–43 indicates medium sensitivity, and >43 indicates high sensitivity. Huang et al. [37] translated the total scale of the MSQ-R-CV to adapt the original three-factor model into a two-factor model: moral strength and responsibility (5 items), and moral burden (4 items). The two-factor model was shown to provide a more robust fit to the empirical data. It is also simpler and more suitable for Chinese nurses in terms of language and culture. Cronbach’s alpha coefficient was 0.820, indicating good internal consistency, similar to that of the English version (0.91). The Cronbach’s alpha coefficient of the scale obtained from this study sample was 0.883; the Cronbach’s alpha of factor 1, “moral responsibility and power”, was 0.893; and the Cronbach’s alpha of factor 2, “moral burden”, was 0.786.

### 2.4. Data Analysis

We used SPSS 25.0 statistical software for data processing and gathered descriptive data by assessing the frequency distribution of variables and parameters. The MSQ-R-CV score for the dependent variable is a continuous variable. Independent sample T-test and one-way analysis of variance were performed if the independent variable was a categorical variable, and correlation analysis was performed if the independent variable was a continuous variable. Statistically significant variables were included in the multiple linear regression equation to test the association between general socio-demographic variables and different dimensions of the WLEIS-C and moral sensitivity. The statistical significance level was set to *p*-value < 0.05.

### 2.5. Ethical Consideration

The researchers obtained the permission of the cooperating units and the informed consent of the participants for this study. Subjects were guaranteed the right to freely choose to participate in or withdraw from the study. All information obtained was only used for this research and handled in an unidentifiable and confidential manner. The research was ethically reviewed and approved by the Institutional Review Board at Central South University (Item No. E202128).

## 3. Results

### 3.1. Sample Description

We collected 467 questionnaires. After screening, 404 questionnaires were included, and the effective response rate was 86.5%. The socio-demographic characteristics of the participants are summarized in Table 1. The average age of the 404 ICU nurses was found to be 30.06 ± 5.52 years. Women accounted for 91.8% of the sample, while men accounted for 8.2%. Married nurses accounted for 62.4%, and those with children accounted for 56.9%. Participants from tertiary hospitals accounted for 90.3%, and those with a bachelor’s degree or above accounted for 80%. The MSQ-R-CV score differed based on variables such as age, marital status, having children, number of years working in an ICU, professional category, and whether the ICU nurses had received ethical training in a hospital or department. All the variables mentioned above were statistically significant. Other details are shown in Table 1.

### 3.2. The Status of Moral Sensitivity of ICU Nurses and Its Relationship with EI

In this study, the average MSQ-R-CV score was 39.41 ± 7.21. The mean ± standard deviation of the WLEIS-C score was 82.39 ± 15.15. The four dimensions and their respective means ± standard deviations were 21.30 ± 4.01 (SEA), 20.19 ± 4.01 (OEA), 19.88 ± 4.37 (ROE), and 21.02 ± 4.11 (UOE). The scores of these four dimensions and the total score of WLEIS-C were significantly correlated with the scores of the MSQ-R-CV scale (*p* < 0.01), as shown in Table 2.

### 3.3. Multiple Linear Regression

Through multiple linear regression, Table 3 shows the variables impacting the moral sensitivity of ICU nurses. MSQ-R-CV scores functioned as dependent variables. Statistically significant variables in Table 1 and Table 2 were independent variables in the regression equation, such as age, marital status, having children, professional category, years working in an ICU, and whether ICU nurses have received ethics training in a hospital or department, along with the scores on various dimensions of the WLEIS-C and the total score of WLEIS-C. Because of the large number of independent variables, to reduce confounding, the step-by-step method was used to screen variables. Finally, the variables that predicted the level of ICU nurses’ moral sensitivity (*p* < 0.05) were as follows: (1) WLEIS-C total score, (2) regulation of emotion, and (3) whether the nurse had children. These variables explained 38.4% of the variation in moral sensitivity among ICU nurses (R^2^ = 0.384, F = 82.967, *p* < 0.001).

According to the standardized coefficient Beta, the total score on the WLEIS-C is directly proportional to the MSQ-R-CV score (B = 0.811, *p* < 0.001), while the regulation of emotion is negatively correlated with the MSQ-R-CV score (B = −0.241, *p* = 0.014). In addition, ICU nurses with children had higher scores on the MSQ-R-CV than those without children (B = −0.093, *p* = 0.02).

## 4. Discussion

The average MSQ-R-CV score of ICU nurses was 39.41 ± 7.21, which indicates a medium level of moral sensitivity. There was a positive correlation between EI and moral sensitivity. The factors influencing moral sensitivity included the regulation of emotion and whether the nurse had children. Unlike previous discussions on EI and moral sensitivity at the theoretical level, this study is the first to conduct quantitative research on the relationship between EI and moral sensitivity.

The average moral sensitivity score of this study was similar to the average moral sensitivity score (40.22 ± 7.08) of general nurses in Hunan Province hospitals [37,40]. The medium level of moral sensitivity of ICU nurses is consistent with other research results [10,41].

As Borhani et al. [10] found no significant relationship between moral sensitivity and moral dilemma in nurses, this may indicate that the frequency of exposure or severity of moral dilemmas has no effect on the levels of moral sensitivity.

We found that EI positively predicted the moral sensitivity of the ICU nurses who participated, which is consistent with a growing body of evidence [29]. Moral sensitivity is based on the recognition of contextual cues to determine moral implications or questions. As a bridge between emotion and rational cognition, EI captures and explains the immediate context and integrates important personal and interpersonal skills—good communication and interaction skills. EI enables nurses to perceive, evaluate, and predict emotional meanings and is often used to inform their deductive reasoning in moral sensitivity. Especially in ambiguous, complex, and challenging or controversial situations [42], EI is more likely to be used to integrate complex clues, leading to comprehensive understanding and plans of action [32]. Emotion is an important driver of cognition, and this integration of emotional and cognitive processing in the brain can serve to promote moral sensitivity and effective decision-making [43]. Therefore, the EI dimension of moral sensitivity must be recognized, not ignored, in nursing.

In the current study, regulation of the emotion dimension of EI was negatively correlated with the moral sensitivity of ICU nurses. This finding is consistent with the ego depletion theory [44]. In moral decision-making, individuals with poor emotional regulation ability consume more emotional resources to empathize with direct stakeholders (for example, relatives of patients, patients, doctors, and even pharmaceutical companies) in decision-making but have fewer emotional resources for monitoring and adjusting their own psychology and behavior. This reduces the amount of energy for self-control, leading to a decrease in their use of emotional regulation strategies. Such individuals are more inclined to morally interpret social information and evoke stronger cognitive and emotional responses when faced with moral dilemmas [45]. Thus, they have higher levels of moral sensitivity. At the same time, low emotional regulation appears to be a negative consequence of high moral sensitivity. Weaver [46] also warns that people tend to interpret moral sensitivity as a desirable quality without critically examining its possible negative consequences accordingly. In fact, moral depression, emotional overload, and loss of self are all associated with high levels of moral sensitivity. However, the research on moral sensitivity lacks attention to this key issue. Consequently, when faced with complex moral decisions or constant moral pressure, ICU nurses with poor emotional regulation may be more likely to break down and experience stronger negative emotions [47]. This makes long-term exposure to other people’s suffering a potential occupational hazard, affecting job burnout, job satisfaction, and nurses’ physical and mental health. Studies (e.g., [48]) have also confirmed that individuals with high moral sensitivity and low emotional regulation have a higher incidence of aggressive behavior, which may disrupt the interpersonal relationships between colleagues and/or doctors and patients. So as not to compromise ICU nurses’ moral sensitivity and empathic abilities, efforts should be devoted to providing timely and effective strategies and resources in emotional regulation. ICU nurses with children had higher scores for moral sensitivity. Their parental role may increase the levels of pressure and responsibilities that they experience, leading to varying degrees of effects on their emotion, cognition, and behavior, which may increase their moral sensitivity.

### 4.1. Impact on Practice

One way to improve moral sensitivity and emotional regulation is to carry out various forms of EI training for clinical ICU nurses (e.g., narrative writing). The review shows that many studies have shown that the intervention plan to cultivate emotional skills in nursing education significantly improves the level of EI in nurses [49], which indicates that EI training is effective [50]. However, training based on emotional development is rare in nursing education programs or healthcare education in general, and the diversity of models and tools used complicates comparisons between studies [49]. Currently recommended training programs guided by the tripartite model of EI focusing on knowledge, ability, and traits have been proven to enable the average healthcare professional to effectively develop EI [51]. A literature review found a trend in research that suggested training is more effective when lectures are avoided and when instruction, practice, and feedback are included [50]. This model suggests the need to personalize EI training and provide accurate feedback on participants’ skills, and that EI can be better trained using active/experiential approaches rather than passive ones [50]. Researchers should further test specific interventions rooted in the EI literature. Through repeated clinical practice, nurses can be guided to learn and acknowledge the significance of their own and others’ emotional experiences and reactions. This may result in an increased acceptance of the importance of EI in clinical settings and more active participation in regulating emotions. In addition, ICU nurses should learn how to trust their emotions in moral decisions and thus promote moral sensitivity from an EI perspective. Nurses with high levels of moral sensitivity, including ICU nurses with children, should be given additional assistance, such as by providing effective emotional regulation strategies, organizing a departmental mental health team to adjust and guide their emotional states, and offering adequate external support. The external support is as follows: (1) Nursing managers should grasp the real psychological changes in nurses through humanized management. This would allow them to help nurses to eliminate negative emotions and reduce the impact of negative emotions in a timely manner by providing an employee psychological assistance program. (2) Hospital management should rationally allocate human resources and increase the allocation of nurses to key clinical departments (such as ICUs) and ensure the flexible arrangement of nursing workloads according to the actual situation to avoid excessive psychological pressure caused by continuous and excessive emotional labor of nurses. (3) Hospitals should create supportive work environments (e.g., reducing or avoiding violence). Studies (e.g., [52]) show that a harmonious and safe working atmosphere can enable nurses to express their emotions more reasonably at work and improve their ability to manage their emotions. Enhancing the support of leaders and colleagues will make nurses more willing to share their feelings sincerely, promote mutual support, and enhance the emotional stability of nurses [53]. (4) Departments should provide more training opportunities for EI to promote effective emotional regulation and management. Educators, psychologists, and clinical experts in nursing should form a team to participate in the preparation and teaching of research courses related to emotion and care in the field of nursing, so as to further improve the capacity of nurses in China to manage the emotional labor that nursing requires.

### 4.2. Study Limitations

This study was conducted in only 10 hospitals in China’s Hunan Province. The generalization of the results to others needs careful consideration. The majority of participants were ICU nurses in tertiary hospitals, and there is a need for a larger and more diverse sample of nurses. Since all data were self-reported, the accuracy of collected information may be limited. Because it was a cross-sectional study, future studies could employ different study methods, such as tracking individuals over time through clinical training programs, to further study EI and measure changes in moral sensitivity before and after interventions.

## 5. Conclusions

This study showed that the moral sensitivity of ICU nurses in Hunan Province was at medium levels. The importance of the EI dimension in ICU nurses’ moral sensitivity should be recognized and attended to, and the specific impact on moral sensitivity of each element in EI should be clarified for practical application. EI training for ICU nurses should be conducted, and effective training methods should be explored. For nurses with high levels of moral sensitivity, more attention should be paid to providing external support, such as guidance on emotional regulation strategies and passive psychological balance by mental health providers. Although there is a growing literature in nursing exploring concepts and ideas related to EI, research on EI and moral sensitivity is still in its infancy. Further research could provide a deeper understanding of the connection between EI and moral sensitivity.

## Figures and Tables

**Table 1 ijerph-19-05132-t001:** Comparison of moral sensitivity scores across general socio-demographic characteristics.

Variables	Scale (*n*, %)	MSQ-R-CV Score Mean ± SD	*p*	t/F
Gender			0.535	−0.626
Male	33 (8.2%)	38.42 ± 9.63		
Female	371 (91.8%)	39.50 ± 6.96		
Age (years)			0.009 *	4.786
≤25	88 (21.8%)	37.58 ± 6.97		
26 ≤ t ≤ 35	252 (62.4%)	39.62 ± 7.08		
≥36	64 (15.8%)	41.09 ± 7.59		
Religious beliefs			0.371	−0.897
Yes	14 (3.5%)	37.71 ± 10.33		
No	390 (96.5%)	39.47 ± 7.08		
Marital status			0.003 *	5.746
Unmarried	138 (34.2%)	37.89 ± 7.08		
Married	252 (62.4%)	40.06 ± 7.10		
Divorced	14 (3.5%)	42.79 ± 7.98		
Having children			<0.001 *	3.747
Yes	230 (56.9%)	40.56 ± 7.19		
No	174 (43.1%)	37.89 ± 6.96		
Hospital level			0.168	1.382
Tertiary hospital	365 (90.3%)	39.57 ± 7.32		
Secondary hospital	39 (9.7%)	37.90 ± 5.84		
Years working in an ICU			0.020 *	3.973
≤1	72 (17.8%)	39.47 ± 7.40		
1 < t ≤ 10	278 (68.8%)	38.91 ± 7.18		
>10	54 (13.4%)	41.91 ± 6.65		
Average night shifts per month			0.100	−1.647
≤8	297 (73.5%)	39.06 ± 6.99		
9≤ t ≤ 20	107 (26.5%)	40.39 ± 7.71		
Education level			0.178	1.736
Junior college	81 (20.0%)	38.93 ± 6.86		
Bachelor’s degree	310 (76.7%)	39.39 ± 7.36		
Master’s or above	13 (3.2%)	42.92 ± 4.35		
Professional category			0.009 *	3.869
Nurse	74 (18.3%)	39.74 ± 7.65		
Senior nurse	176 (43.6%)	38.17 ± 7.09		
Supervising nurse	133 (32.9%)	40.41 ± 7.15		
Deputy director of nursing or above	21 (5.2%)	42.33 ± 5.05		
Work intensity			0.904	0.101
More relaxed and average	39 (9.6%)	39.56 ± 6.95		
Tiring	247 (61.1%)	39.51 ± 6.84		
Very tiring	118 (29.2%)	39.16 ± 8.03		
Having received ethics courses at university			0.305	1.028
Yes	216 (53.5%)	39.75 ± 7.17		
No	188 (46.5%)	39.02 ± 7.25		
Having received ethics training in a hospital or their department			0.046 *	2.001
Yes	137 (33.9%)	40.41 ± 6.93		
No	267 (66.1%)	38.90 ± 7.30		

Note: * indicates that the difference is statistically significant. Abbreviation: MSQ-R-CV, the Moral Sensitivity Questionnaire-Revised Version into Chinese.

**Table 2 ijerph-19-05132-t002:** The total score of MSQ-R-CV and WLEIS-C scale, and the correlation between each dimension.

Variables	MSQ-R-CV Score	SEA	ROE	UOE	OEA	WLEIS-C Score
**MSQ-R-CV Score**	1					
**SEA**	0.586 **	1				
**ROE**	0.506 **	0.783 **	1			
**UOE**	0.583 **	0.836 **	0.781 **	1		
**OEA**	0.542 **	0.779 **	0.778 **	0.783 **	1	
**WLEIS-C score**	0.603 **	0.924 **	0.914 **	0.925 **	0.908 **	1

Note: ** *p* < 0.01 (two-tailed), the correlation is significant. Abbreviations: MSQ-R-CV: the Moral Sensitivity Questionnaire-Revised Version into Chinese. WLEIS-C: Wong and Law Emotional Intelligence Scale, Chinese version. SEA: self-emotion appraisal. ROE: regulation of emotion. UOE: use of emotion. OEA: others’ emotion appraisal.

**Table 3 ijerph-19-05132-t003:** Variables related to nurses’ ethical sensitivity (stepwise multiple linear regression, N = 404).

Variables	Unstandardized Coefficient		Standardization Coefficient Beta	t	*p*	R^2^	F	*p*
	B	Standard Error						
						0.384	82.967	<0.001
(Constant)	17.486	1.921		9.104	<0.001			
WLEIS-C	0.385	0.047	0.811	8.256	<0.001			
ROE	−0.397	0.161	−0.241	−2.470	0.014			
Having children	−1.351	0.581	−0.093	−2.327	0.020			

Abbreviation: WLEIS-C: Wong and Law Emotional Intelligence Scale Version into Chinese. ROE: regulation of emotion.

## Data Availability

The data presented in this study are available upon request from the corresponding author. The data are not publicly available due to restrictions, e.g., privacy or ethical.

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
