# Peer review of "Moral Sensitivity and Emotional Intelligence in Intensive Care Unit Nurses"

_ijerph, 2022, doi:10.3390/ijerph19095132_

Round 1
Reviewer 1 Report
Thank you for sharing this interesting insight on moral sensitivity and emotional intelligence among ICU nurses, it brings empirical soundness to their correlation.
The results have been presented clearly and concisely, and the study brings novel insights into the relationship between emotional intelligence and moral sensitivity.
However, improvement would be helpful if their correlation could be presented graphically. Furthermore, conclusions might be improved by clarifying how each element of EI should be clarified at the practical level in praxis.
Reviewer 2 Report
Thank you for allowing me to review this manuscript. The present study presents an analysis of the correlation between emotional inteligence and moral sensitivity in critical care nurses. The topis is of interest and the paper is good. In general terms, the manuscript is well-written.
I would lik to point out some suggestions of improvement points:
Abstract:
- The abstract has no methods information and I would recommend to specify the sections included.
- In line 6, I do not understand the sentence "it is necessary then to explain..." what do you mean?
- Please replace all p=0.000 from the manuscript and abstract for p<0.001
- In line 12, Authors say "the importance of EI in ICU..." this first part of the sentence is not supported by results. Conclusions should be derived from results.
- Keywords from abstract should not include abbreviations, please include full terms
Citation style: please see author guidelines as Vancouver is not the recommended one.
The introduction is a bit difficult to follow, I would suggest to change the information included in order to make it easier to understand for all potential readers. Moreover, for me it's difficult to understand the link between all the explained parts and the aim of the objective. The backgrpund and all information that authors provide is difficult to follow
Lines 72-73: Sentence "studies have demonstrated...." references are lacking. Please include them.
Methods:
- Sample size calculation is not adequate for this kind of studies. The approach proposed by authors is to evaluate the psychometric properties from a questionnaire. Please redo the sample size calculation
- sentence from lines 103-104 belongs to results and it's already there, So please delete it from here
Discussion:
- I would suggest that in the text ROE is used as full term, not the abbreviation to facilitate the reading
- In implications for practice authors say that nurses should be trained on emotional inteligence but do not provide evidence of its effectiveness or guidance on how to do it. I would recommend to add information regarding these issues
